# Dynamic lattice distortions driven by surface trapping in semiconductor nanocrystals

Burak Guzelturk [1,2,15,16 ✉], Benjamin L. Cotts [1,16], Dipti Jasrasaria [3,16], John P. Philbin [3,16], David A. Hanifi[1,3], Brent A. Koscher[3,4], Arunima D. Balan[3,4], Ethan Curling[3], Marc Zajac[1], Suji Park[2], Nuri Yazdani[2,5], Clara Nyby[6], Vladislav Kamysbayev[7], Stefan Fischer[1], Zach Nett[3], Xiaozhe Shen [8], Michael E. Kozina[8], Ming-Fu Lin [8], Alexander H. Reid [8], Stephen P. Weathersby[8], Richard D. Schaller [9,10], Vanessa Wood [5], Xijie Wang [8], Jennifer A. Dionne [1], Dmitri V. Talapin [7,9], A. Paul Alivisatos[3,4,11,12], Alberto Salleo [1], Eran Rabani[3,4,13] & Aaron M. Lindenberg [1,2,6,14 ✉]

Nonradiative processes limit optoelectronic functionality of nanocrystals and curb their device performance. Nevertheless, the dynamic structural origins of nonradiative relaxations in such materials are not understood. Here, femtosecond electron diffraction measurements corroborated by atomistic simulations uncover transient lattice deformations accompanying radiationless electronic processes in colloidal semiconductor nanocrystals. Investigation of the excitation energy dependence in a core/shell system shows that hot carriers created by a photon energy considerably larger than the bandgap induce structural distortions at nanocrystal surfaces on few picosecond timescales associated with the localization of trapped holes. On the other hand, carriers created by a photon energy close to the bandgap of the core in the same system result in transient lattice heating that occurs on a much longer 200 picosecond timescale, dominated by an Auger heating mechanism. Elucidation of the structural deformations associated with the surface trapping of hot holes provides atomic-scale insights into the mechanisms deteriorating optoelectronic performance and a pathway towards minimizing these losses in nanocrystal devices.

[1] Department of Materials Science and Engineering, Stanford University, Stanford, CA, USA. [2] Stanford Institute for Materials and Energy Sciences, SLAC National Accelerator Laboratory, Menlo Park, CA, USA. [3] Department of Chemistry, University of California, Berkeley, CA, USA. [4] Materials Sciences Division, Lawrence Berkeley National Laboratory, Berkeley, CA, USA. [5] Department of Information Technology and Electrical Engineering, ETH Zurich, Zurich, Switzerland. [6] The PULSE Institute for Ultrafast Energy Science, SLAC National Accelerator Laboratory, Menlo Park, CA, USA. [7] Department of Chemistry and James Franck Institute, University of Chicago, Chicago, IL, USA. [8] SLAC National Accelerator Laboratory, Menlo Park, CA, USA. [9] Center for Nanoscale Materials, Argonne National Laboratory, Lemont, IL, USA. [10] Department of Chemistry, Northwestern University, Evanston, IL, USA. [11] Department of Materials Science and Engineering, University of California, Berkeley, CA, USA. [12] Kavli Energy NanoScience Institute, Berkeley, CA, USA. [13] The Sackler Center for Computational Molecular and Materials Science, Tel Aviv University, Tel Aviv, Israel. [14] Department of Photon Science, Stanford University and SLAC National Accelerator Laboratory, Menlo Park, CA, USA. [15]Present address: X-ray Science Division, Argonne National Laboratory, Lemont, IL, USA. [16]These authors contributed equally: Burak Guzelturk, Benjamin L. Cotts, Dipti Jasrasaria, John P. Philbin. ✉email: burakg@anl.gov; aaronl@stanford.edu

Nonradiative relaxation processes in materials represent fundamental loss mechanisms, which set performance limits in electronics, optoelectronics, and photocatalysis. Nonradiative relaxation events become further critical in devices of quantum-confined materials such as nanocrystals (NCs) and nanowires due to their high surface-to-volume ratios. As such, intensive research efforts have been focused on identifying non-radiative losses and the means to circumvent them in nanomaterials[1,2]. Among these, colloidal semiconductor NCs have attracted significant technological interest due to their appealing optoelectronic properties[3,4], which are tunable via shape, size, composition, and surface chemistry[1,5–7]. Today, state of the art NCs can reach near-unity radiative efficiencies[8] but these are typically measured under moderately weak excitation conditions. In applications, such as lasers[9], photodetectors[10], multiexciton-harvesting solar cells[11], and electrically pumped LEDs[12], NCs are commonly exposed to high energy and/or high intensity excitation conditions, where nonradiative relaxation rapidly escalates.

Previously, high photon energy excitation of NCs has been shown to cause increased blinking[13], reduced photoluminescence quantum yields[14,15], and increased photoionization[16]. These observations have suggested that hot carriers in NCs can lead to severe charge trapping, increasing nonradiative losses. In addition, Auger recombination becomes dominant in NCs that have more than one exciton[17]. A hot carrier is created at the expense of an annihilated exciton via an Auger process, hence substantially curbing the performance of NC lasers[18] and LEDs[19]. Although earlier works focused on identifying optical signatures associated with such nonradiative processes in NCs[20,21], more recent works have begun to point to the fundamental role of dynamic structural fluctuations interrelated with nonradiative relaxation in NCs[22–29]. In this context, neutron scattering measurements corroborated by molecular dynamics (MD) simulations[22] and correlative transmission electron microscopy (TEM) studies[27] have indicated that NC surfaces are mechanically soft, and thus may accelerate the nonradiative relaxation process. Nevertheless, such structural deformations associated with nonradiative relaxations in photoexcited NCs have remained elusive to date and have never been directly probed on ultrafast timescales.

Here, we perform femtosecond electron diffraction[30] measurements on prototypical cadmium chalcogenide colloidal NCs to directly probe the atomic scale responses following photoexcitation. We investigate the effects of excitation photon energy on the transient atomic responses in thin film samples of core/shell and core-only NCs. Studying the core/shell sample under different excitation energies enables selective excitation of the core vs. the shell; thus, we decouple the effects of NC surfaces on nonradiative relaxations. We find that Auger recombination dominates the transient heating of the core/shell NCs when multiexcitons are generated in the core by photons with energies close to the bandgap of the core. The transient heating response is corroborated by MD simulations. On the other hand, we unveil that localized disordering is induced in addition to transient heating when multiexcitons are generated predominantly in the shell by photons with energies much larger than the bandgap of the shell. These localized structural deformations arise from localization of hot holes at NC surfaces forming surface small polarons (Fig. 1a). Kinetic models considering these nonradiative relaxations capture the experimentally measured dynamics well. Furthermore, measurements on a core-only sample are presented, which indicate that hole trapping happens under both excitation photon energies yet with different temporal dynamics implying the presence of an energy barrier for surface hot hole trapping in this system.

## Results

### Ultrafast electron diffraction of CdSe/CdS nanocrystals.
Figure 1a schematically depicts the femtosecond electron diffraction measurements performed in a transmission geometry, where we monitor the diffraction from NC thin films deposited on TEM grids as a function of pump-probe delay. Figure 1b shows the radially integrated diffraction intensity in the absence of optical pump $I_0(Q)$, where $Q$ is the scattering vector. The sample in Fig. 1b is a CdSe/CdS core/shell NC with a shell thickness of eight monolayers (ML) (see "Methods" for sample details and Supplementary Fig. 1 for TEM images). Figure 1b also shows the transient change in the diffraction intensity $\Delta I(Q,t)$ measured at a pump-probe delay of $t \approx 500$ ps when the sample is excited at 510 nm. The intensity of all diffraction peaks, labeled from Q1 to Q7 (see corresponding reciprocal planes in Supplementary Table 1), decreases transiently, while the intensity in the diffuse scattering region (in-between the peaks) increases. The relative loss of diffraction peak intensity implies that the NCs become transiently disordered after photoexcitation.

### Excitation with low photon energy in the core/shell nanocrystal.
We first discuss measurements of the core/shell sample when excited at 510 nm. Note that 510 nm excitation predominantly excites the CdSe core (see Supplementary Fig. 2). Figure 2a shows the relative diffraction intensity ($I(t)/I_0$) represented at four different diffraction peaks at an excitation fluence of 2.1 mJ cm$^{-2}$. Changes in $I(t)/I_0$ become progressively larger for higher $Q$ peaks. This $Q$-dependence resembles a transient heating response known as the Debye–Waller (DW) effect, where diffraction peak intensities decrease as the material heats up due to increased mean squared atomic displacements ($\langle \Delta u(t)^2 \rangle$). In a DW model under harmonic assumption, $I(t)/I_0$ can be related to $\langle \Delta u(t)^2 \rangle$ via $-\ln(I(t)/I_0) = \frac{Q^2}{3} \langle \Delta u(t)^2 \rangle$ (Supplementary Section D)[31]. To check experimental agreement with the DW model, we plot $-\ln(I(t)/I_0)$ as a function of $Q^2$ in Fig. 2b at $t = 1000$ ps. A linear relationship with zero intercept holds for all fluences studied between 1.5 and 3.1 mJ cm$^{-2}$ (see also Supplementary Fig. 3). This observation implies that the time-dependent structural response of the core/shell NCs when excited by 510 nm primarily originates from transient heating. We estimate $\langle \Delta u(t)^2 \rangle$ as shown in the inset of Fig. 2b, which scales linearly as a function of fluence indicating that the absorbed energy density per NC also increases linearly with the excitation fluence.

To gain better insight into the structural deformations occurring in response to photoexcitation in these NCs, we calculate a differential atomic pair distribution function (PDF) $\Delta G(r,t)$ revealing transient changes in the atomic pair correlations[32,33] (see Supplementary Section C). In a wurtzite CdSe (or CdS), the first atomic pair correlation peak is at 2.5 Å, which corresponds to the first nearest neighbor Cd–Se (or S) bond distance[34]. Higher order correlations, including those corresponding to the distances across the $a$- and $c$-axis of the unit cell (Fig. 2c, inset) at 4.1 and 7.1 Å, respectively, are also observed. At each correlation distance, we observe a transient dip at the peak center and a rise on each side (Fig. 2c). This indicates that atomic pair correlations are transiently broadened, as expected from transient heating of the NCs[33]. To further validate this, we perform MD simulations calculating $\Delta G(r,\Delta T)$ resulting from a static temperature increase of $\Delta T$ (Supplementary Section D). The simulated $\Delta G(r,\Delta T = 14$ K$)$ ($T_0 = 300$ K) is plotted in Fig. 2c, showing good agreement with the experimental $\Delta G(r,t)$, further supporting our conclusion that the lattice response in this case is dominated by transient heating.

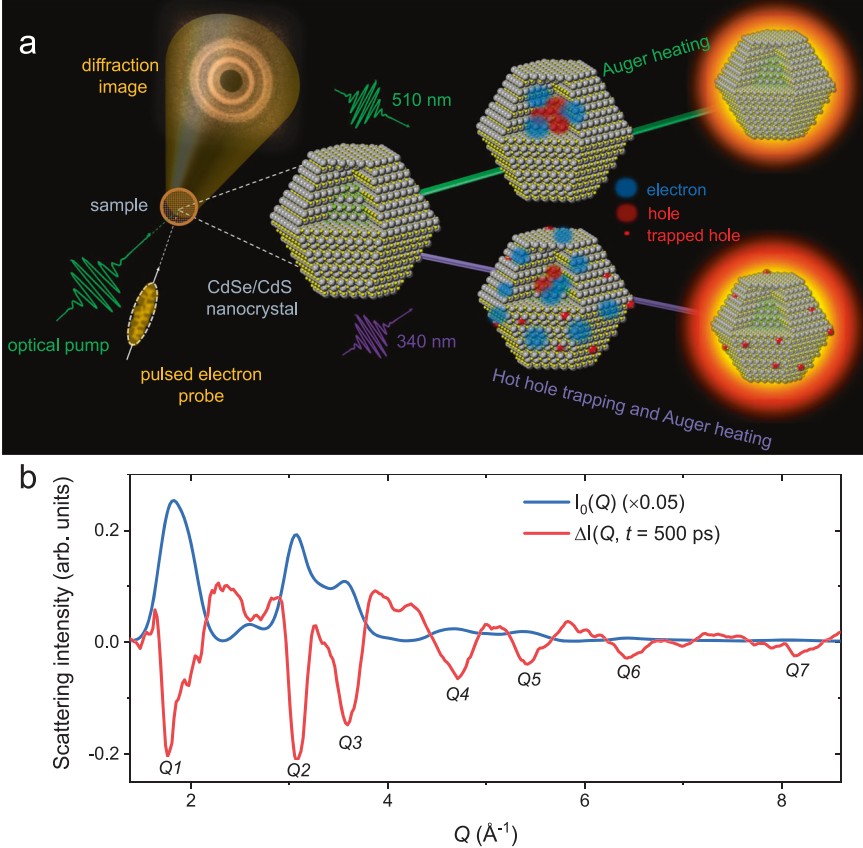

**Fig. 1 Femtosecond electron diffraction on colloidal nanocrystals. a** Schematic demonstration of the femtosecond electron diffraction, where we perform optical pump/electron-beam diffraction probe experiments on colloidal nanocrystals (NCs) deposited on TEM grids. We observe dynamic lattice heating and localized surface disordering associated with nonradiative relaxations in NCs. In a CdSe/CdS core/shell NC, Auger heating dominates the response with 510 nm (green colored) excitation, while hot carrier surface trapping prevails with 340 nm (purple colored) excitation. Electrons and holes are denoted by blue and red colors, respectively. **b** $I_0(Q)$ is the radially integrated diffraction intensity (solid blue) in the absence of optical pump in a CdSe/CdS core/shell NC. $\Delta I(Q,t = 500 \text{ ps})$ is the transient change in the diffraction intensity (solid red) measured at a pump/probe delay of 500 ps with 510 nm excitation under a fluence of 2.1 mJ cm$^{-2}$. Different diffraction peaks are labeled as Q1–Q7.

We convert $\langle \Delta u(t)^2 \rangle$ into lattice temperature changes $\Delta T(t)$ by considering the DW factors calculated by our MD simulations (Supplementary Fig. 5), which are also in good agreement with prior reports[31]. Figure 2d shows $\Delta T(t)$ along with $\Delta E(t)$, which is the energy transferred to the NC lattice (Supplementary Section E). $\Delta T$ reaches a quasi-equilibrium at ca. 13 K under an excitation fluence of 2.1 mJ cm$^{-2}$ consistent with the MD simulation above. We fit $\Delta T(t)$ phenomenologically by a single exponential function, which gives lifetimes on the order of 200 ps (Supplementary Fig. 13). This implies an exceptionally gradual heating of the NCs, which could be explained either by (1) a bottleneck during the course of cascaded energy transfer from hot carriers into optical phonons and then into acoustic phonons[35], or (2) an Auger heating mechanism[36]. A bottleneck between hot carriers and optical phonons is not considered, as prior reports have established that this coupling is rather fast (≤1 ps) in colloidal NCs[37]. However, a bottleneck may exist in the down-conversion of the emitted optical phonons into acoustic ones, where the acoustic phonons are more prominent in the DW response because of their larger contribution to mean squared atomic displacements (Supplementary Fig. 6)[38]. Previously, a bottleneck between optical and acoustic phonons has been alluded to in lead-halide perovskites through a hot phonon bottleneck effect[35] due to the efficient up-conversion of acoustic phonons into the optical ones. To check on this mechanism, we calculate phonon density of states and corresponding phonon relaxation lifetimes via MD

simulations. We find that the phonon relaxation lifetimes, which are dictated by the anharmonic phonon–phonon interactions, are typically < 1 ps (Fig. 2e). This lifetime reaches ~10 ps only for the smallest energy acoustic phonons (<0.5 THz), but this is still an order of magnitude faster than the heating of the NCs. Therefore, we rule out hot phonon bottleneck as the primary mechanism underlying the slow heating response.

Next, we check the hypothesis of an Auger heating mechanism, which arises due to the generation of hot carriers at delayed times via Auger recombination (Fig. 2f). We propose a simple kinetic model that considers the Auger heating mechanism, where the rate of heating is approximately equal to the rate of Auger recombination (Supplementary Section F). We consider the dependence of the Auger recombination rate[39] on the average number of excitons per NC ($\langle N \rangle$) as $\frac{N(N-1)}{2} \frac{1}{\tau_{AR}}$, where $\tau_{AR}$ is the biexciton Auger lifetime. $\tau_{AR} \approx 625$ ps in the core/shell NC (Supplementary Fig. 11). We calculate $\langle N \rangle$ based on the absorption cross-section at 510 nm, which is ca. 10 for a fluence of 1 mJ cm$^{-2}$ and scales linearly with the fluence (inset of Fig. 2b). The Auger heating model (solid lines in Fig. 2d) completely captures the experimental dynamics including both the time scale as well as the signal amplitude. This strongly implies that Auger heating is the predominant mechanism contributing to the transient heating of the core/shell NCs when many excitons are generated near the band edge in the core of the NCs.

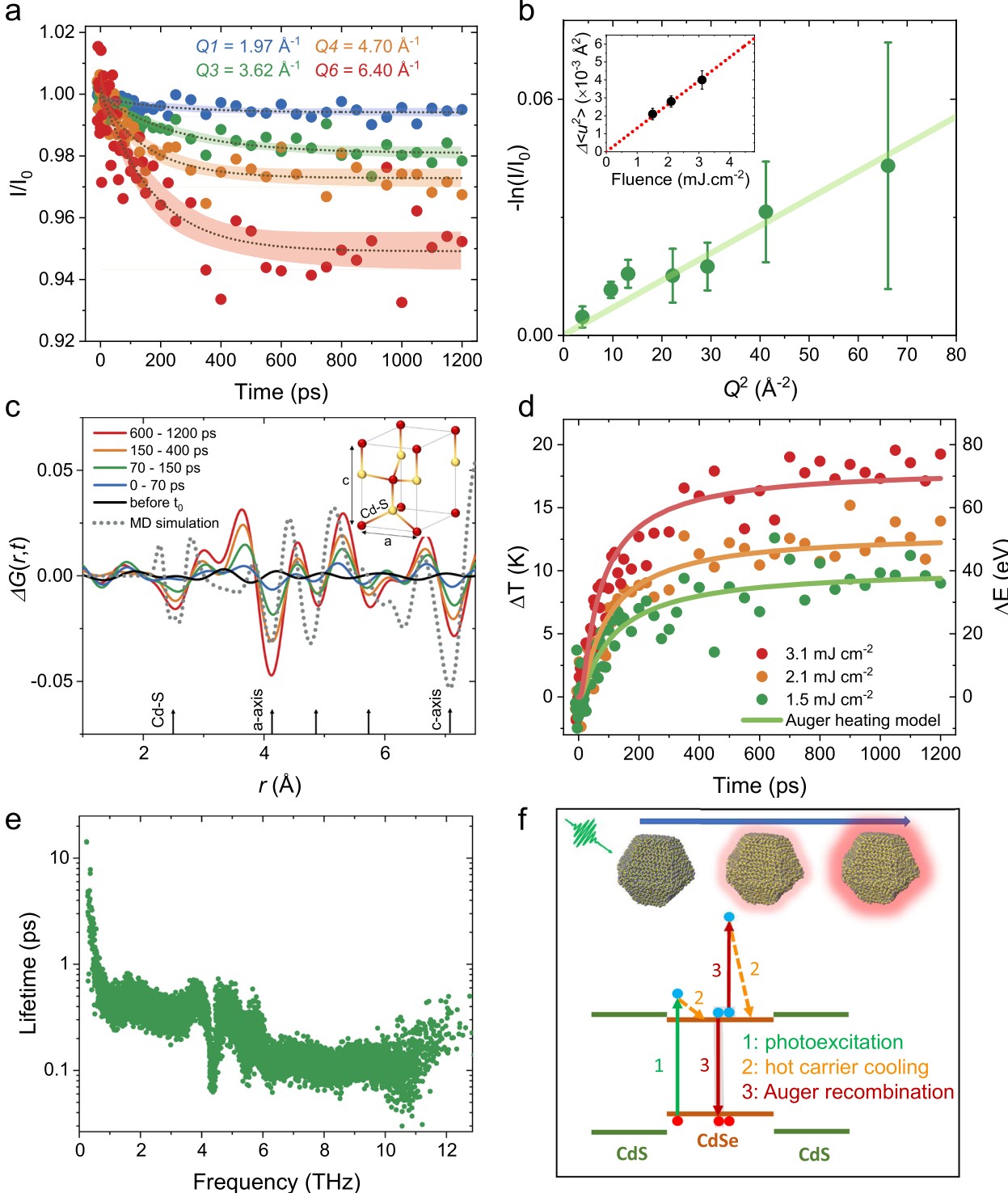

**Fig. 2 Low photon energy excitation of the core/shell NCs. a** $I(t)/I_0$ is the relative diffraction intensity as a function of pump/probe delay shown for four different diffraction peaks labeled by $Q1$, $Q3$, $Q4$, and $Q6$. **b** $-\ln(I(t)/I_0)$ plotted as a function of $Q^2$ at $t = 1000$ ps, which shows a linear response confirming that the transient effect arises from a Debye–Waller (DW) effect. The inset shows the calculated induced mean squared atomic displacements $<\Delta u(t)^2>$ for three different fluences. The error bars show standard error. **c** $\Delta G(r,t)$ is the differential atomic pair distribution function measured with respect to $G(r)$ of unexcited NCs at chosen time delays of $t$ (solid lines). Molecular dynamics (MD) simulated $\Delta G(r, \Delta T = 14$ K$)$ is also shown for static temperature increase of 14 K (over 300 K) in the same NCs (dashed line). The inset shows a wurtzite CdS unit cell with the three nearest neighbor distances (Cd–S, $a$- and $c$-axis) marked. **d** $\Delta T(t)$ shows the transient increase in lattice temperature along with $\Delta E(t)$, which shows the transient energy transferred to the lattice. The solid lines are the fits based on Auger heating model. **e** Phonon relaxation lifetimes at 300 K calculated by MD simulations. **f** Schematic showing the relevant nonradiative relaxation channels modeled here. Top schematic indicates the transient heating process of the NCs dominated by Auger heating.

**Excitation with high photon energy in the core/shell nanocrystal**. We now discuss 340 nm excitation of the core/shell NCs. Supplementary Fig. 14 shows $I(t)/I_0$ measured at four different diffraction peaks, which shows that the transient structural responses occur significantly faster under this excitation (~20 ps). Figure 3a plots $-\ln(I(t)/I_0)$ over $Q^2$ at $t = 200$ ps. Although low $Q$

diffraction peaks (Q1–Q5) show a linear-like response among themselves (solid line in Fig. 3a), higher $Q$ peaks (Q6 and Q7) strongly deviate from this linearity. This deviation happens consistently for all studied fluences between 2.5 and 6.0 mJ cm$^{-2}$ (see Supplementary Fig. 15). Therefore, this suggests additional transient deformations occurring in these NCs concurrently with

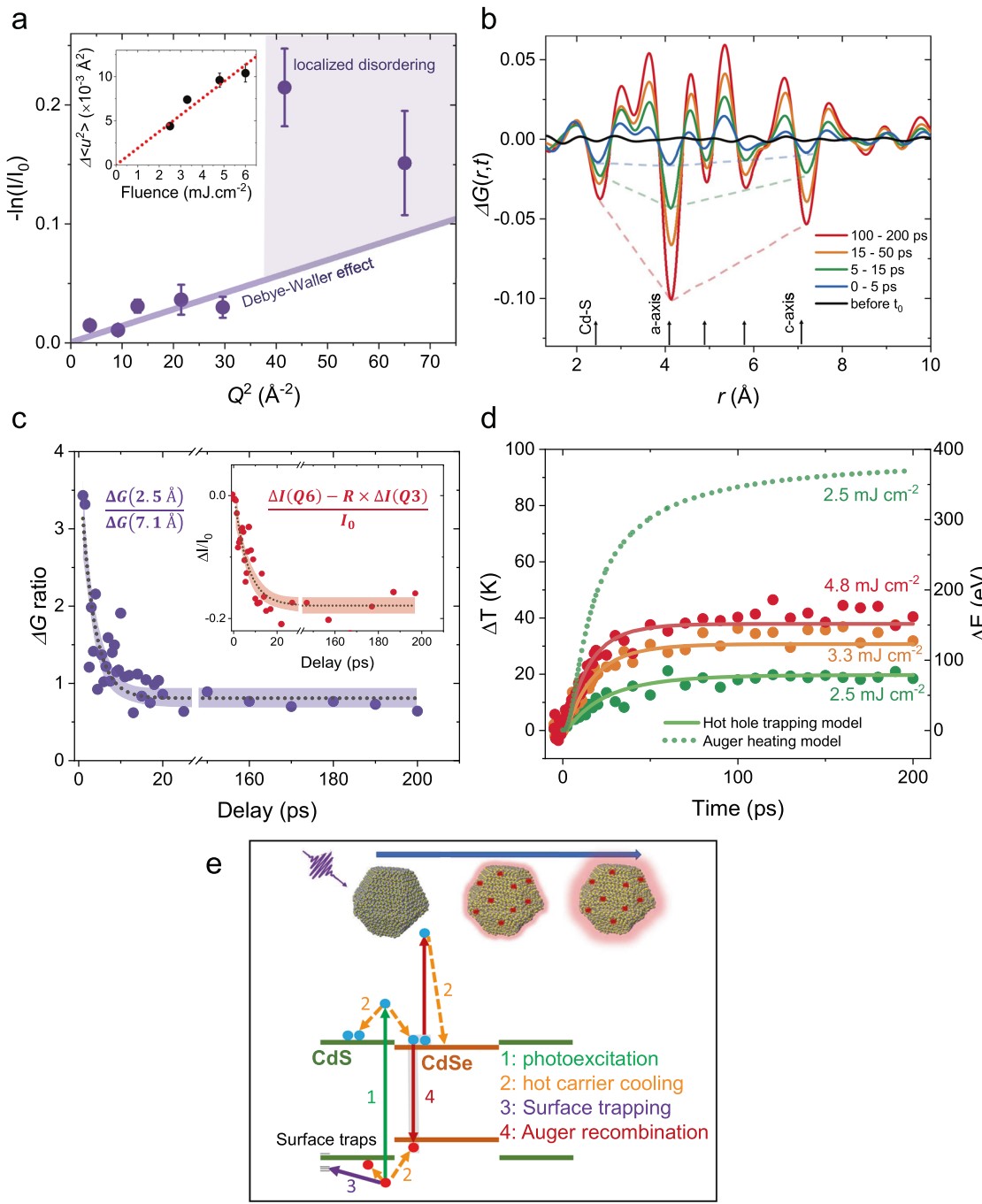

**Fig. 3 High photon energy excitation of the core/shell NCs. a** $-\ln(I(t)/I_0)$ plotted as a function of $Q^2$, which strongly deviates from a linear response. Although the first five $Q$ peaks are linear among themselves (solid line is linear fit), higher order $Q$ peaks ($Q^2 > 40$ Å$^{-2}$) deviate from linearity. This indicates a transient lattice effect that cannot be explained by a DW effect alone. The error bars show standard error. **b** Differential atomic pair distribution function $\Delta G(r,t)$ at chosen $t$ ranges. Dashed lines highlight the dips at 2.5, 4.1, and 7.1 Å. **c** Ratio of differential atomic pair correlation at the first nearest neighbor with respect to that of the wurtzite $c$-axis ($\Delta G(2.5$ Å$,t)/\Delta G(7.1$ Å$,t)$) to disentangle the localized disorder dynamics. The inset shows the dynamics obtained by subtracting $I(t)/I_0$ at $Q3$ from that of $Q6$. $R$ is a proportionality factor inferred from change in Debye–Waller factor from $Q3$ to $Q6$. **d** Experimental $\Delta T(t)$ and $\Delta E(t)$ in the case of 340 nm excitation. The solid lines are the fits based on fast trapping model. **e** Schematic showing the relevant nonradiative relaxation channels modeled here. Top schematic also indicates the transient localized lattice disordering associated with surface hole trapping in addition to transient heating by hot carrier relaxation.

the transient heating. Particularly, the deviation at high $Q$ implies that the induced disorder is linked to the formation of short length scale, localized lattice deformations. This can be understood with the fact that a diffraction peak at $Q$ probes a real space order of $2\pi/Q$. For example, the $Q6$ peak at $6.4$ Å$^{-1}$ probes ~1 Å in real space. Thus, strong disordering of the high $Q$ peaks implies disordering at the smallest spatial length scale, which we denote here as a localized structural disordering[40].

To understand the nature of these localized structural deformations, we perform differential atomic PDF analysis. Figure 3b shows $\Delta G(r,t)$ under an excitation fluence of 4.8 mJ cm$^{-2}$. An investigation of the transient dips at different atomic correlations indicates that at early delay times the first nearest neighbor correlation at $r = 2.5$ Å is more affected than all other correlations (see dashed lines in Fig. 3b). This observation implies that the localized disordering has the largest impact on the first atomic correlation peak, which is consistent with prior work in other materials with highly localized disorder[41]. Also, this implies that the localized disordering proceeds faster than the transient heating, which is supported by a comparison of the dynamics of $\Delta G(r,t)$ at different $r$. Supplementary Fig. 16 shows the time evolution of the correlation loss amplitude at correlations with large $r$ of 4.1 and 7.1 Å, which exhibit the same time constant of $20 \pm 1$ ps. Thus, the effect for large $r$ correlations is the same and its response is dominated by the transient heating. On the other hand, $\Delta G(r,t)$ at $r = 2.5$ Å exhibits considerably faster kinetics with a time constant of 11 ps. In this case, local disordering and transient heating both contribute to the dynamics together. To decouple the dynamics associated with the formation of localized deformations, Fig. 3c shows $\Delta G(2.5$ Å$,t)$ normalized by $\Delta G(7.1$ Å$,t)$, where the normalization effectively removes the heating dynamics. After the normalization, we estimate a time constant of ~3.5 ps directly linked with the time scale for the formation of localized lattice deformations.

In addition, diffraction intensity changes ($I(t)/I_0$) measured at different $Q$ peaks (Supplementary Fig. 14) reveal the same time scale for the formation of localized deformation. While $I(t)/I_0$ at lower $Q$ peaks exhibit a time constant around ~23 ps dictated by the transient heating, the $Q6$ peak shows a time constant of ~10 ps. To extract the dynamics of the localized deformations in this case, we subtract $I(t)/I_0$ measured at a low $Q$ peak ($Q3$), scaled to the linear DW value at $Q6$, from that of the experimental data at $Q6$, obtaining a time constant of 6 ps (see inset of Fig. 3c). This faster time scale at high $Q$ is consistent with the comparative PDF analysis above. The inset of Fig. 3c also shows that these localized deformations do not relax within the measured time window of 200 ps and hence are long lived. However, they relax within a 2.7 ms time window, which is the excitation repetition rate in this experiment. Additional longer-lived distortions may also be induced which are not probed in these time-dependent measurements.

Hot carriers in the NCs can be trapped via localization of the carriers at the NC surfaces at picosecond timescales[21,42–44] causing broad defect emissions[42], reduced photoluminescence quantum yields[15] and increased blinking[13,45]. Ab initio calculations have also suggested that trapping may be linked with the dynamic atomic fluctuations of the poorly passivated surface chalcogen atoms[44,46–48]. Our observations here indicate that the localized lattice deformations are formed on picosecond timescales under 340 nm excitation, where hot carriers are dominantly excited in the shell region close to the surface of the NCs. In this context, we hypothesize that the localized atomic deformations arise from dynamic reconstruction of the NC surfaces as hot carriers localize at poorly passivated surface atoms forming surface small polarons[29]. To validate this hypothesis, we investigate $\Delta T(t)$ (Fig. 3d), which is estimated from the lower $Q$ peaks exhibiting DW-like response. Note here that, the $\langle N \rangle$ is ~100 at 2.5 mJ cm$^{-2}$ estimated from the absorption cross-section at 340 nm. We find that the Auger heating model substantially overestimates the amplitude of $\Delta T(t)$ by a factor of 4 (dashed line in Fig. 3d) although absorbed energy density by the NCs scales linearly (Fig. 3a, inset). This implies that Auger heating must be suppressed in this case. Consistent with the hypothesis, Auger recombination has been observed to be repressed in the NCs with surface trapped holes[49] as the trapping leads to spatial separation of the carriers in a NC. We extend our kinetic model to account for the suppression of the Auger heating due to competition with fast hot carrier surface trapping (Supplementary Section F). We apply the time constant for formation of the localized deformations as the time scale for hot carrier trapping. This model (solid lines in Fig. 3d) agrees well with the experimental $\Delta T(t)$, which strongly implies that the transient structural response in the case of 340 nm excitation is dominated by the localized surface carrier trapping (Fig. 3E).

Both types of hot carriers are created in the shell region by 340 nm. Based on only this information, we cannot differentiate which carrier dominates the trapping process. However, with 510 nm excitation, hot carriers are created predominantly in the core region. Due to the band alignment between CdSe and CdS, electrons can be delocalized throughout the whole NC, while holes are localized to the core[39]. No significant localized deformations are observed with 510 nm excitation, which implies that the delocalized hot electrons cannot be the main cause of the surface trapping. Thus, hot holes must govern the formation of localized deformations as they trap at the NC surfaces. This is consistent with prior theoretical work in cadmium chalcogenide NCs, which have suggested that the main carrier that leads to trapping is the hole due to poorly passivated surface chalcogen atoms[23,29,44]. The spatial extent of the localized distortions arising from hot hole trapping can be estimated from the dynamic structural information (Supplementary Section H). Comparing the relative disorder introduced to the first and second atomic correlation peaks under 510 nm (Fig. 2c) vs. 340 nm (Fig. 3b) permits approximation of these localized distortions under the assumption that a single hot hole localizes at a single unit cell at the surface. We find the amplitude of the localized distortion to be ~0.15 Å per trapped hole, a key input to future theoretical studies of surface trapped charge, or surface small polaron, transport in nanomaterials. Furthermore, close examinations of the diffraction peaks associated with the localized disorder (e.g., $Q6$) reveal larger weighting of the $l$ component of the ($hkl$) Miller indices (see Supplementary Table 1). This implies that the local distortion induced by the small polaron within the unit cell favorably involves deformations with a significant component along the $c$-axis of the wurtzite unit cell of the NCs (see Supplementary Section I).

**Excitation of the core-only nanocrystal**. We also measure the transient structural responses in a CdSe core-only NC sample, which is the same size core used in the core/shell sample. Figure 4a shows $-\ln(I(t)/I_0)$ as a function of $Q^2$ for both 340 and 510 nm excitations. We observe that the localized lattice deformations, evidenced by an increased loss at high $Q$ diffraction peaks, occur under both excitation cases, while the effect is much more pronounced for the 340 nm excitation. Differential atomic PDF analysis in Fig. 4b shows the dynamics associated with the localized disordering. The localized distortions emerge with a $6.9 \pm 3.1$ ps time constant with 340 nm excitation, consistent with the core/shell sample under the same excitation condition. On the other hand, the localized lattice disordering proceeds with a slow

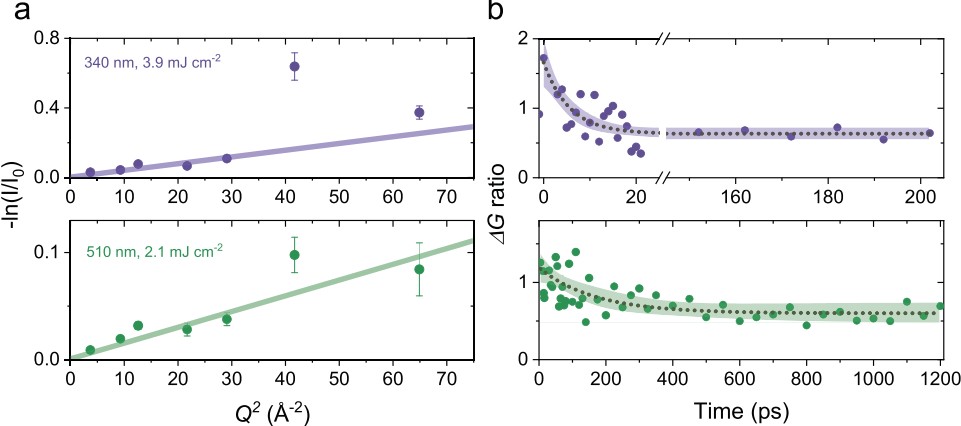

**Fig. 4 Excitation of the core-only CdSe NCs. a** $-\ln(I(t)/I_0)$ plotted as a function of $Q^2$. Top panel is for 340 nm excitation, bottom panel is for 510 nm excitation. In both cases, the experimental signal deviates from a linear response due to the presence of localized lattice disordering. The error bars show standard error. **b** Ratio of differential atomic pair correlation at the first nearest neighbor with respect to that of the wurtzite $c$-axis ($\Delta G(2.5\,\text{Å},t)/\Delta G$ $(7.1\,\text{Å},t)$). Top panel is for 340 nm and bottom panel is for 510 nm. The localized lattice disorder, hence localized charge trapping, happens with a time constant of 6.9 and 167 ps in the cases of 340 and 510 nm excitations, respectively. The color shaded region around dotted curve shows the exponential fit and 95% confidence interval. In the case of 510 nm, the slow response arises from Auger recombination-induced hot hole generation, which leads to trapping at later times.

time constant of $167 \pm 70$ ps with the 510 nm excitation. This indicates that hot holes generated by 510 nm are not energetic enough to cause localized surface trapping, while those generated by 340 nm are. In the case of 510 nm excitation, Auger recombination leads to the generation of energetic hot holes at later times which underlies the slower formation of localized deformations. This observation implies that there is a finite energy barrier for the formation of localized surface hole traps. Considering the excess energy of the hot holes in the core-only and core/shell NCs, we estimate that the energy barrier for hole trapping is >0.1 and <0.36 eV. In the case of core/shell sample under 510 nm excitation, the absence of strong localized disorder signal (Fig. 2b and Supplementary Fig. 3) implies that hole trapping is not a predominant channel. We think the important reason behind this is that Auger recombination favors hot electron generation over hot hole generation in this core/shell system. Our theoretical estimate indicates a 3:1 ratio for hot electron to hot hole generation, hence a suppressed hot hole population due to Auger process at later times (see Supplementary Section G). On the other hand, in a core-only sample the hot electron and hot hole Auger channels are roughly equivalent.

## Discussion

Femtosecond electron diffraction applied to colloidal semiconductor NCs directly visualizes nonradiative relaxations occurring in photoexcited semiconductor NCs in real time with an atomic-scale resolution. With this, we uncover the dynamical structural responses associated with the formation of localized surface charge traps and Auger recombination. We show that hot holes with excess kinetic energy induce short range atomic deformations extending ~0.15 Å as these carriers localize at surface trapping sites and form surface small polarons. Our results indicate that excitation energy management in NCs by minimizing the excess energy of hot hole is crucial to suppress non-radiative losses associated with surface trapping. As such, high energy excitation in NC lasers and energetic hole injection in LEDs should be avoided to minimize undesired surface trapping, important for wider technological deployment of semiconductor NCs in applications.

## Methods

**Femtosecond electron diffraction experiments**. UED experiments were conducted at the SLAC National Accelerator Laboratory MeV-UED instrument, a part of the LCLS User Facility. The experimental setup and our analysis approach have been detailed before[33]. A multipass Ti:sapphire laser (800 nm, 60 fs, 360 Hz) is used to drive both an optical parametric amplifier to create a tunable energy ultrafast optical pump and to excite a photocathode to drive the electron bunch pulses. The electron bunch probe pulses are accelerated to 3.7 MeV to achieve ~200 fs pulse widths with 50 fC charge per pulse. Diffracted electrons were detected using an EMCCD via a red phosphor. Time zero was calibrated for using either thin single-crystal silicon or bismuth samples.

High-quality samples of core-only CdSe and eight ML core-shell CdSe/CdS were used from the same batches detailed in ref. [8] and drop-cast onto TEM grids. Full synthetic and characterization details for the quantum dot stock solutions can be found within ref. [8], with details in Supplementary Section A. Samples were imaged before and after measurements at the MeV-UED facility to confirm that no damage took place during measurement. For results in Figs. 2–4, time-resolved measurements are repeated and averaged over >10 different scans.

**Time-resolved photoluminescence**. To characterize biexciton Auger lifetime, we measured the core/shell sample under 510 nm excitation. For this, we used a 35 fs amplified Ti:sapphire laser system with a 2 kHz repetition rate. The output of the laser is converted into 510 nm using an optical parametric amplifier. The NC solution (optical density of 0.1 at 510 nm) was placed in a 1-mm-thick quart cuvette and excited with varying excitation fluences. The excitation beam size was 496 μm in diameter. The sample was kept stirring throughout the measurement with the help of a small magnet. To capture the photoluminescence decay curves, we used a streak camera (Hamamatsu) providing an instrument response function full-width at half-maximum of 30 ps.

**Molecular dynamics simulations**. MD simulations were performed on CdSe/CdS core/shell NCs using the LAMMPS code[50] and a previously implemented interatomic pair potential parameterized for CdSe and CdS[51] (see details in Supplementary Section D). The static temperature differential atomic PDF (Fig. 2c) was calculated using the eight ML core/shell NC. Radial distribution functions were computed directly using LAMMPS from equilibrium MD trajectories at 314 and 300 K and then transformed to the PDFs, smoothed, and subtracted to obtain $\Delta G(r, \Delta T = 14\,\text{K})$. Mean square atomic displacements (MSD) for the eight ML core/shell NC were computed from equilibrium MD trajectories at temperatures ranging from 150 to 500 K. A linear relationship between temperature and change in MSD was found and used to estimate experimental transient lattice temperatures.

The phonon density of states was computed for a four ML core/shell NC. The structure was minimized using the conjugate descent algorithm implemented in LAMMPS. This configuration was used to compute the mass-weighted Hessian, which was diagonalized to obtain the phonon frequencies and modes. The lifetimes for each of these phonon modes (Fig. 2e) were computed within a linear response formalism[52]. Equilibrium MD simulations were used to compute the velocity autocorrelation function for each mode, which was then used to compute the

Langevin friction kernel via a numerical Laplace transform and obtain the phonon lifetime (see details in Supplementary Section D).

**Kinetic models.** The kinetic models consist of sets of coupled differential equations (Eqs. S1–S12 in Supplementary Section F). These equations were solved using the Gillespie algorithm[53], which uses trajectories with varying time steps to solve classical master equations. The trajectories were initialized with electron–hole pair populations according to the Poisson distribution with the average number of electron–hole pairs consistent with the absorption cross-section and the optical pump fluence. A linear relationship between the amount of electronic energy lost via phonons and the temperature increase of the lattice is assumed. The Auger recombination lifetime was fit to simultaneously reproduce the time dynamics of the time-resolved photoluminescence and ultrafast electron diffraction data in Fig. 2d, whereas hot hole surface trapping is taken into account for the data in Fig. 3d (see Supplementary Section F).

## Data availability
Source data are provided with this paper. Other data sets are available from B.G. and A.M.L. upon reasonable request.

## Code availability
Analysis codes used for analyzing UED data are available from B.G., B.L.C., and D.J. upon reasonable request.

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

## Acknowledgements

This work is primarily part of the "Photonics at Thermodynamic Limits" Energy Frontier Research Center funded by the US Department of Energy, Office of Science, Office of Basic Energy Sciences under Award Number DE-SC0019140. MeV-UED is operated as part of the Linac Coherent Light Source at the SLAC National Accelerator Laboratory, supported by the US Department of Energy, Office of Science, Office of Basic Energy Sciences under Contract No. DE-AC02-76SF00515. This work was performed, in part, at the Center for Nanoscale Materials, a US Department of Energy Office of Science User Facility, and supported by the US Department of Energy, Office of Science, under Contract No. DE-AC02-06CH11357. Part of this work was performed at the Stanford Nano Shared Facilities (SNSF), supported by the National Science Foundation under award ECCS-1542152. R.D.S. and D.V.T. acknowledge support from NSF DMREF Program under awards DMR–1629361, DMR–1629601, and DMR–1629383. N.Y. and V. W. acknowledge funding from Swiss National Science Foundation from the Quantum Sciences and Technology NCCR. M.Z. and S.P. acknowledge support from the Department of Energy, Office of Science, Basic Energy Sciences, Materials Sciences and Engineering Division, under Contract DE-AC02-76SF00515. D.J. acknowledges the support of the Computational Science Graduate Fellowship from the US Department of Energy under Grant No. DE-SC0019323.

## Author contributions

A.M.L. and B.G. conceived the experiment. B.G. and B.L.C. led the UED experimental team consisting of B.G., B.L.C., D.A.H., B.A.K., A.D.B., M.Z., S.P., N.Y., C.N., V.K. and S.F. SLAC UED team consisting of X.S., M.E.K., M.F.L, A.H.R., S.P.W. and X.W. assisted the experiments. B.G. and B.L.C. performed data analysis of the experimental UED data. B.A.K., Z.N. and E.C. synthesized the nanocrystals. B.L.C., B.G., D.A.H., B.A.K. and E.C. prepared the UED samples. B.G., B.L.C., A.M.L., D.J., J.P.P., E.R. and A.S. interpreted the data. B.G. and R.D.S. performed transient photoluminescence measurements. D.J. performed the MD simulations. J.P.P. and B.G. developed kinetic models. B.G. and B.L.C. wrote the paper with contributions from all authors.

## Competing interests

The authors declare no competing interests.
