## [Peer Review File · Nature Communications]

Reviewer #1 (Remarks to the Author):

In their manuscript, B. Guzelturk et al. report detailed results concerning the non-radiative relaxation processes involved in photoexcited colloidal CdSe/CdS core/shell nanocrystals. Mainly combining time-resolved electron diffraction and molecular dynamics simulations, the authors are able to identify these processes for two different photoexcitation regimes. For an optical excitation with photons having an energy close to the CdSe bandgap, only Auger recombinations are involved. In the case of photons with an energy much larger than the CdSe and CdS bandgaps, distortions of the lattice occur and hot holes are trapped at the CdS shell surface.

The results are very clearly presented and discussed. The conclusions are also well supported by a large amount of data and are of great interest for a wide range of applications of colloidal nanocrystals. The data are also provided by numerous and sometimes original methods (electron diffraction and molecular dynamics simulations but also photoluminescence decay measurements and predictions of kinetic models) that give new insights at very short time scales. For all these reasons, I believe that the paper could be published in Nature Communications. However, some points have to be clarified before publication:

1) For peaks Q1 to Q5, a linear response of $-\ln(I(t)/I_0)$ is always evidenced. For Q6 and Q7, a very large deviation is observed in the high energy excitation regime. However, the error bars for Q6 and Q7 are very large in Figure 2b (low energy regime). Could the authors comment these 2 results and justify why a linear response can be ruled out?

2) Could the authors also justify:

- why the transition between Q5 ($\sim 30 \text{ \AA}^{-2}$) and Q6 ($\sim 40 \text{ \AA}^{-2}$) is so abrupt in figure 3a?
- why a decrease is observed between Q6 and Q7?

3) For the data in figure 4b, the authors provide a decay constant of 6 ps and 150 ps without precision. However, the points at short delays exhibit a high dispersion. Do the dotted curves correspond to fits or are only a guide for the eye?

4) The authors suggest (end of page 6) that there is a link between these results and the extensive literature concerning blinking of single colloidal nanocrystals. Is this correct? In this case, how can the authors be sure that the distortion “relax within a 2.7 ms time window, which is the excitation repetition rate in this experiment” since blinking at longer time scales have been observed by many authors, even with thick shell nanocrystals?

Reviewer #2 (Remarks to the Author):

The authors of this manuscript present novel mixed optical/electron diffraction experiments to probe the nature of photo-relaxation and trapping in nanocrystals. Using these experiments, and interpretations aided by molecular dynamics simulations, they make valuable contributions to the field of non radiative relaxation in nanocrystals, and so I find the work to be of high quality and a good fit for Nature Communications. I really like that the authors seem to be observing polaron formation during hole trapping in some cases--very cool.

I would like the authors to be more precise in their language around one of the most important

aspects of this paper. It is not exactly controversial that "normal" Cd-chalcogenide nanoparticles have hole traps on the surface, and that those traps are accessible to holes even when the particles are excited near the band gap. If hole trapping isn't observed, either the particles are special--they have been designed to eliminate the surface hole traps--or the experiment isn't sensitive to hole trapping.

In several places this paper implies that hole trapping doesn't occur for the low energy excitation: The abstract says "carriers created by a photon energy close to the bandgap result in transient lattice heating that occurs on a much longer 200 ps timescale, governed by an Auger heating mechanism."

The picture in figure 1 shows that hole trapping is nonexistent at low energies.

Lines 232-234:

excitations. We observe that the localized lattice deformations, evidenced by an increased loss at high q diffraction peaks, occur under both excitation cases, while the effect is much more pronounced for the 340 nm excitation.

Lines 238-239

...This indicates that hot holes generated by 510 nm are not energetic enough to cause localized surface trapping, while those generated by 340 nm are.

These statements appear to be in contradiction.

My interpretation is that you see lattice distortions owing to hole trapping for the core only case for both low and high energy photons, but the heating signal overlaps with it at low energies. I do not disagree with the assignment of the nuclear distortion to hole trapping, I disagree with assigning its absence to the absence of hole trapping.

If the authors are claiming that hole trapping does not occur for "core-only" nanocrystals when excited near the bandgap, that needs an explanation. I think that what the authors are saying instead is that the core-shell particles heat at low energies but do not trap, and that the core only particles trap and heat at low energies. Both particles trap at high energies. So either I am misunderstanding something or the authors need to be more clear about the fact that the two classes of particles behave differently at low energies.

Reviewer #3 (Remarks to the Author):

The authors apply femtosecond electron diffraction measurements, coupled with molecular dynamics modelling, to investigate the atomic scale effects of photoexcitation at different energies in CdSe core-only and CdSe/CdS core/shell nanocrystals. They find that energies close to the band gap lead to Auger-recombination-mediated transient heating of the NC. Higher energy excitations lead instead to localised disordering, when the hot hole localises at a surface trap. Assuming single-hot-hole occupancy

per surface unit cell, they estimate localised distortions of the order of 0.15 Å per trapped hole. This estimate represents an invaluable information for performing accurate theoretical modelling of these systems. This is, however, not the only significant result of this study. Other noteworthy features are:

the application of ultrafast electron diffraction measurements to extract detailed information about transient atomic-scale deformations on these systems and the conclusion that localised atomic deformations arise from dynamic reconstruction of the NC surface as hot holes localise at poorly passivated surface atoms. This conclusion is supported by both the results displayed in Fig.3 and by the kinetic model developed by the authors, and sheds new light on the trapping dynamics and its effects at the structural atomic level.

The paper is well presented and logically structured. The methodology is sound and the detail provided should be sufficient for the work to be reproduced. The results presented are both timely and very interesting for the whole community.

I therefore recommend publication.

There is only one aspect that I did not quite understand, and would like the authors to clarify in the revised manuscript:

If the interpretation of the origin of the data in Fig.4a is correct [the slower formation of localised deformations for 510 nm excitation is due to the hot hole created by AR], why don't we see a deviation from linear behaviour for higher order Q peaks ($Q^2 > 40 \text{ \AA}^{-2}$) in Fig.2b as well? Indeed, given that AR creates hot holes in core-shell systems as well (and with the same excess energy as in core-only NCs), they would have sufficient energy to access the CdS surface, and create a delayed localized lattice deformation, as in the case of core-only excitation at the same energy. As this deformation is long lived it should be captured in the plot of $-\ln[I(t)/I_0]$ at 1000 ps (i.e., in Fig.2b).

If this is the case, the statement that "excitation energy management in nanocrystals by minimizing the excess energy of hot hole is crucial to suppress nonradiative losses associated with surface trapping" does not seem to be correct, as it is the high excitation fluence (not the energy) needed for lasers that, by creating more than one exciton in the NC, makes it possible for the AR-generated hot hole to access surface traps.

Marco Califano

Response to Referees' comments

We would like to thank each referee for their careful reading of our manuscript and thoughtful comments which have helped us to further enrich our manuscript and we are grateful for this. Below we have addressed all the points raised by the referees and provide point by point responses along with the modifications made in the revised manuscript.

Reviewer #1 (Remarks to the Author):

In their manuscript, B. Guzelturk et al. report detailed results concerning the non-radiative relaxation processes involved in photoexcited colloidal CdSe/CdS core/shell nanocrystals. Mainly combining time-resolved electron diffraction and molecular dynamics simulations, the authors are able to identify these processes for two different photoexcitation regimes. For an optical excitation with photons having an energy close to the CdSe bandgap, only Auger recombinations are involved. In the case of photons with an energy much larger than the CdSe and CdS bandgaps, distortions of the lattice occur and hot holes are trapped at the CdS shell surface.

The results are very clearly presented and discussed. The conclusions are also well supported by a large amount of data and are of great interest for a wide range of applications of colloidal nanocrystals. The data are also provided by numerous and sometimes original methods (electron diffraction and molecular dynamics simulations but also photoluminescence decay measurements and predictions of kinetic models) that give new insights at very short time scales. For all these reasons, I believe that the paper could be published in Nature Communications.

Author Response:

We thank the reviewer for the appreciation of our work and positive remarks. We address all the points raised by the Referee below.

However, some points have to be clarified before publication:

1) For peaks Q1 to Q5, a linear response of $-\ln(I(t)/I_0)$ is always evidenced. For Q6 and Q7, a very large deviation is observed in the high energy excitation regime. However, the error bars for Q6 and Q7 are very large in Figure 2b (low energy regime). Could the authors comment these 2 results and justify why a linear response can be ruled out?

Author Response:

We thank the Reviewer for this question. The error bars (defined as the standard error) become larger for signals measured at higher Q peaks as they have smaller overall diffraction intensities. Nevertheless, we observe statistically significant behavior for the two different photoexcitation cases in terms of $-\ln(I(t)/I_0)$ vs. Q^2 appearing consistently across a range of fluences. Below we append Figure S3 and Figure S15 from the supplementary information as Figure R1 and Figure R2 which show the responses under different excitation fluences at 510 nm and 340 nm, respectively.

Figure R1 shows that a linear relationship is consistently preserved in the CdSe/CdS core/shell sample for all three fluences measured under 510 nm excitation. In the case of 340 nm excitation of the same sample, Figure R2 shows four different fluence cases, which consistently exhibits a non-

linear behavior particularly associated with the deviation at high Q peaks. Also, a linear fit (dotted black curves in Figure R2) cannot capture the behavior in the case of 340 nm.

Figure R1. $-\ln(I(t)/I_0)$ plotted as a function of Q^2 for different excitation fluences under 510 nm excitation. All fluence cases show a linear response confirming that the transient effect arises from a Debye-Waller (DW) effect.

Figure R2. $-\ln(I(t)/I_0)$ as a function of Q^2 under 340 nm excitation at 200 ps for fluences of 6.0, 4.8, 3.3, and 2.5 mJ cm^{-2} . Each excitation fluence displays a linear response at low Q ($<30 \text{ \AA}^{-2}$) and additional localized disorder at high Q ($>40 \text{ \AA}^{-2}$). Dotted lines show the linear fit to the whole range. Colored lines show the linear fit to the lowest five Q peaks.

2) Could the authors also justify:

- why the transition between Q_5 ($\sim 30 \text{ \AA}^{-2}$) and Q_6 ($\sim 40 \text{ \AA}^{-2}$) is so abrupt in figure 3a?

- why a decrease is observed between Q_6 and Q_7 ?

Author Response:

We thank the Referee for pointing out this important question. To first order, the strong deviation in the transient signal for the high Q peaks indicates that the nanocrystals are undergoing an additional localized structural disordering in addition to the simple homogenous heating response under 340 nm excitation. This is reflected in the pair distribution function analysis real space transformation of this data that reveals additional early time short-range disorder in Figure 3c in comparison to the lower energy excitation. Furthermore, in the revised manuscript we now provide additional insights into why there is a sharp onset to the Q-dependent response of the transient localized disordering as below:

The spatial extent and local configuration of the surface hole polaron depend on the local symmetry, bonding and orbital structure. Thus, it is complicated to precisely estimate the true nature of the small surface polarons. Prior theory works have predicted that charges in various nanocrystals (e.g., CdS or metal oxides) may be localized to form polarons lying within particular atomic planes or atomic sites (e.g. R. Cline et al. JPCL **9**, 3532, 2018 and J. Carey et al. JPCC **122**, 27540, 2018). This would favor distortions along particular crystallographic directions. In this picture, we believe that the deviation starting with the Q6 peak can be understood by considering the relevant diffraction peaks underlying each of the first six peaks Q1-Q6. Supplementary Table 1 is shown below showing our indexing of the Miller indices (*hkl*) of these peaks.

Supplementary Table 1: Major contributing reciprocal planes for each Q peak cited in the main text (Fig. 1b). Due to the sample-detector distance chosen to maximize q range and the overlapping contributions from CdS and CdSe, the peaks are not separately distinguished. Some minor peaks are omitted.

	Q1	Q2	Q3	Q4	Q5	Q6	Q7
Major contributing reciprocal planes	(100) (002) (101)	(110)	(112)	(203) (210) (211)	(213) (302)	(215) (116) (222)	Unassigned*

*beyond reported range in ICDD tables [PDFs 000-041-1049 and 00-008-0459][†]

One notes that Q6 is the only peak involving large *l* indices (e.g., (215) and (116)), corresponding to lattice planes oriented perpendicular to the *c*-axis of the wurtzite unit cell of the nanocrystals. Thus, the deviation in peak Q6 from the linear response associated with the isotropic heating is indicative of unit cell distortions occurring favorably along the *c*-axis, which are amplified by the large *l* in the structure factor (given by $F_{hkl} = \sum_n f_n e^{2\pi i(hx_n + ky_n + lz_n)}$ with diffracted intensity proportional to the modulus squared of this quantity). In fact, in this model one may potentially understand in a self-consistent way why it is that peaks Q1 and Q3 lie above the linear fit while Q2 lies below (as is the case for all fluences shown above in Fig. R2) since Q2 in particular corresponds to a set of planes with *l*=0, aligned parallel to the *c*-axis with structure factor insensitive to distortions along *z/c*. In summary, the sharp deviation of peak Q6 from the linear Debye-Waller-like response can be understood as indicative of the development of short-range distortions likely involving significant *c*-axis-oriented displacements. Higher *Q*-resolution measurements in the future could potentially shed further light on the atomic distortions involved.

These discussions above are reflected in the main text on page 8 and Supplementary information. “Furthermore, close examinations of the diffraction peaks associated with the localized disorder (e.g., Q6) reveal larger weighting of the *l* component of the (*hkl*) Miller indices (see Supplementary Table 1). This implies that the local distortion induced by the small polaron within the unit cell

favorably involves deformations with significant component along the c-axis of the wurtzite unit cell of the nanocrystals (see Supplementary Section I).”

3) For the data in figure 4b, the authors provide a decay constant of 6 ps and 150 ps without precision. However, the points at short delays exhibit a high dispersion. Do the dotted curves correspond to fits or are only a guide for the eye?

Author Response:

We now provide error bars to the fits used in Figure 4b which are 167 ± 70 ps and 6.9 ± 3.1 ps on page 8. The dotted curves are the fits with their 95% confidence intervals. The revised caption now notes this.

4) The authors suggest (end of page 6) that there is a link between these results and the extensive literature concerning blinking of single colloidal nanocrystals. Is this correct? In this case, how can the authors be sure that the distortion “relax within a 2.7 ms time window, which is the excitation repetition rate in this experiment” since blinking at longer time scales have been observed by many authors, even with thick shell nanocrystals?

Author Response:

To clarify this, we note that the distortions probed in our pump-probe measurements are fully reversible since we are always comparing the structure relative to the structure at negative times (where the pump arrives after the probe) averaged over many pulses. Thus, all transient signals shown in Figures 2-4 relax within the 2.7 ms time window. If they didn't, we would not be able to resolve them when averaging over many pulses. Nevertheless, the referee is correct that we cannot rule out the presence of additional long-lived average deviations from the equilibrium structure associated with longer relaxation times. These would show up as a static offset in the diffraction patterns with laser on relative to the equilibrium structure.

On the other hand, we think that a direct structural observation of polaronic hole traps has close relevance to the blinking properties of the nanocrystals as trapping is considered as one of the main reasons for blinking. Also, our observation is strongly linked to prior observations of increased blinking in nanocrystals upon higher photon excitation energy (Ref. 13). One significant distinction, however, is that our observation is based on an intense excitation regime ($\langle N \rangle \gg 1$) totally different than that ($\langle N \rangle \ll 1$) of the conventional single dot blinking studies. Thus, the timescales of recovery may not be directly translated into the conventional blinking experiments. Also, given the limited signal to noise ratio associated with the structural signatures of trapping, we cannot completely rule out if a few hot holes still remain trapped within the nanocrystals.

We now clarify this above point at the bottom of page 6 noting that additional longer-lived distortions may also be present, not probed in this experiment.

Reviewer #2 (Remarks to the Author):

The authors of this manuscript present novel mixed optical/electron diffraction experiments to probe the nature of photo-relaxation and trapping in nanocrystals. Using these experiments, and interpretations aided by molecular dynamics simulations, they make valuable contributions to the field of non radiative relaxation in nanocrystals, and so I find the work to be of high quality and a good fit for Nature Communications. I really like that the authors seem to be observing polaron formation during hole trapping in some cases--very cool.

Author Response:

We thank the Referee for the positive remarks and appreciation of our work.

I would like the authors to be more precise in their language around one of the most important aspects of this paper. It is not exactly controversial that "normal" Cd-chalcogenide nanoparticles have hole traps on the surface, and that those traps are accessible to holes even when the particles are excited near the band gap. If hole trapping isn't observed, either the particles are special--they have been designed to eliminate the surface hole traps--or the experiment isn't sensitive to hole trapping.

In several places this paper implies that hole trapping doesn't occur for the low energy excitation: The abstract says "carriers created by a photon energy close to the bandgap result in transient lattice heating that occurs on a much longer 200 ps timescale, governed by an Auger heating mechanism."

The picture in figure 1 shows that hole trapping is nonexistent at low energies.

Lines 232-234:

We observe that the localized lattice deformations, evidenced by an increased loss at high-diffraction peaks, occur under both excitation cases, while the effect is much more pronounced for the 340 nm excitation.

Lines 238-239

...This indicates that hot holes generated by 510 nm are not energetic enough to cause localized surface trapping, while those generated by 340 nm are.

These statements appear to be in contradiction.

My interpretation is that you see lattice distortions owing to hole trapping for the core only case for both low and high energy photons, but the heating signal overlaps with it at low energies. I do not disagree with the assignment of the nuclear distortion to hole trapping, I disagree with assigning its absence to the absence of hole trapping.

If the authors are claiming that hole trapping does not occur for "core-only" nanocrystals when excited near the bandgap, that needs an explanation. I think that what the authors are saying instead is that the core-shell particles heat at low energies but do not trap, and that the core only particles trap and heat at low energies. Both particles trap at high energies. So either I am misunderstanding something or the authors need to be more clear about the fact that the two

classes of particles behave differently at low energies.

Author Response:

We thank the referee for highlighting this important question. First, localized lattice disorder associated with hole trapping occurs in the core-only sample under both excitation conditions as shown by Figure 4. Thus, we agree with the reviewer, and are not claiming that hole trapping does not occur for “core-only” nanocrystals when excited near the bandgap. However, the time-scales for the formation of localized disordering, hence hot hole trapping, is significantly different for 340 nm (< 10 ps) vs. 510 nm (>100 ps). This observation implies that the excess energy of the hot holes plays an important role in the trapping process. We hypothesize that the slow response in the case of 510 nm is due to the hot hole trapping facilitated mainly by the hot holes generated by Auger recombination. Thus, we suggest that an energy barrier exists for the hot hole trapping through surface polaron formation. Such excitation energy dependence for hole trapping has been alluded in the Cd-chalcogenide nanocrystals before by prior steady state photoluminescence and transient absorption studies (Kambhampati et al. Chem. Phys. 446, 92 (2015), Htoon et al. Small 10, 2892 (2010)) but the nature of such dynamic traps was not well understood as transient structural information was missing to date.

We also agree with the reviewer that the absence of signal associated with localized disorder does not imply the absence of any hole trapping. Given the signal to noise ratio associated with the transient structural signals, we would be unable to distinguish a small number of trapped holes in the presence of the background heating response. However, weak or absent signal indicates that the trapping is not the dominant kinetic channel in this excitation condition (510 nm and core/shell sample). Therefore, we modify our claim for the case of 510 nm excitation of the core/shell sample by stating that the hot hole trapping is not a prominent channel as we cannot rule out minor amount of trapping beneath our signal noise floor.

The ambiguities highlighted by the referee are addressed in the revised manuscript as per the following.

We modified the abstract and the associated parts in the main text (page 3 and 8) to better distinguish the conclusions regarding the trapping responses of the core/shell and core-only samples.

Reviewer #3 (Remarks to the Author):

The authors apply femtosecond electron diffraction measurements, coupled with molecular dynamics modelling, to investigate the atomic scale effects of photoexcitation at different energies in CdSe core-only and CdSe/CdS core/shell nanocrystals. They find that energies close to the band gap lead to Auger-recombination-mediated transient heating of the NC. Higher energy excitations lead instead to localised disordering, when the hot hole localises at a surface trap. Assuming single-hot-hole occupancy per surface unit cell, they estimate localised distortions of the order of 0.15 Å per trapped hole. This estimate represents an invaluable information for performing accurate theoretical modelling of these systems. This is, however, not the only significant result of this study.

Other noteworthy features are:

the application of ultrafast electron diffraction measurements to extract detailed information about transient atomic-scale deformations on these systems and the conclusion that localised atomic deformations arise from dynamic reconstruction of the NC surface as hot holes localise at poorly passivated surface atoms. This conclusion is supported by both the results displayed in Fig.3 and by the kinetic model developed by the authors, and sheds new light on the trapping dynamics and its effects at the structural atomic level.

The paper is well presented and logically structured. The methodology is sound and the detail provided should be sufficient for the work to be reproduced. The results presented are both timely and very interesting for the whole community. I therefore recommend publication.

Author Response:

We thank the Referee for the positive remarks and appreciation of our work.

There is only one aspect that I did not quite understand, and would like the authors to clarify in the revised manuscript:

If the interpretation of the origin of the data in Fig.4a is correct [the slower formation of localised deformations for 510 nm excitation is due to the hot hole created by AR], why don't we see a deviation from linear behaviour for higher order Q peaks ($Q^2 > 40 \text{ \AA}^{-2}$) in Fig.2b as well? Indeed, given that AR creates hot holes in core-shell systems as well (and with the same excess energy as in core-only NCs), they would have sufficient energy to access the CdS surface, and create a delayed localized lattice deformation, as in the case of core-only excitation at the same energy. As this deformation is long lived it should be captured in the plot of $-\ln[I(t)/I_0]$ at 1000 ps (i.e., in Fig.2b). If this is the case, the statement that "excitation energy management in nanocrystals by minimizing the excess energy of hot hole is crucial to suppress nonradiative losses associated with surface trapping" does not seem to be correct, as it is the high excitation fluence (not the energy) needed for lasers that, by creating more than one exciton in the NC, makes it possible for the AR-generated hot hole to access surface traps.

Marco Califano

Author Response:

We thank the referee for this insightful question. To begin, we cannot completely rule out trapping in the core/shell samples under 510 nm excitation given the signal to noise ratio and the underlying background signal associated with the heating response. Nevertheless, the weak (or absent) trapping signal with lower energy excitation (510 nm) in the core/shell sample strongly indicates that the hot hole trapping is not a predominant channel in this excitation condition and sample. To explain this point, we now highlight two important mechanisms.

First, we show that Auger recombination favors hot electron formation over the hot hole formation in these core/shell heterostructures. Our new calculations (now appended in the revised Supplementary Information Section G) show that, in a comparably sized core/shell quantum dot (CdSe core with a diameter of 3.8 nm and a 4 ML CdS shell), the biexciton Auger recombination lifetime that produces a high energy electron has a lifetime of ~ 280 ps whereas the channel that

produces a high energy hole has a lifetime of ~ 967 ps. This difference in lifetimes simply means that $\sim 75\%$ of the hot carriers generated at later times via Auger recombination would be electrons. As a result, this selectivity in Auger recombination is expected to curb the population of hot holes in the core/shell sample, hence leading to weaker trapping associated with hot holes at later times. Interestingly, the electron and hole Auger recombination lifetime channels are approximately equal in CdSe core-only quantum dots.

Second, we think that the direct photoexcitation of a large number of hot holes with 340 nm excitation can amplify the trapping process through Coulomb repulsion between hot holes in the shell region at early times just after photoexcitation, forcing the hot holes to interact with the surface regions of the nanocrystals more. On the other hand, smaller numbers of hot holes generated by Auger recombination process at later times in the core region would not be affected as much by such a process. Plus, these hot holes generated by Auger in the core region would be quickly thermalized and relocalized to the core region in strong competition with the surface trapping process. Therefore, we think that the core/shell samples provide an increased immunity against hot hole trapping under the lower excitation energy conditions by avoiding direct excitation of the shell. We think this insight would be useful to exploit nanocrystals in applications such as lasing for better performance.

We now present our new calculations on the different Auger recombination lifetimes for the electron and hole channels in the core/shell quantum dots in Supplementary Information Section G. We also revised the relevant discussion in the main text on page 8 accordingly with the discussion above.

Reviewer #1 (Remarks to the Author):

The authors responded satisfactorily to the issues raised by the referees. I therefore recommend publication of the manuscript in Nature Communications.

Reviewer #2 (Remarks to the Author):

The authors have addressed my criticism with the interpretation of the results. I recommend the paper be accepted.

Joel Eaves

Reviewer #3 (Remarks to the Author):

The revised manuscript addresses my concern. I therefore recommend publication